# The Landscape of Secondary Genetic Rearrangements in Pediatric Patients with B-Cell Acute Lymphoblastic Leukemia with t(12;21)

**DOI:** 10.3390/cells12030357

**Published:** 2023-01-18

**Authors:** Agnieszka Kaczmarska, Justyna Derebas, Michalina Pinkosz, Maciej Niedźwiecki, Monika Lejman

**Affiliations:** 1Student Scientific Society of Independent Laboratory of Genetic Diagnostics, Medical University of Lublin, A. Gębali 6, 20-093 Lublin, Poland; 2Department of Pediatrics, Hematology and Oncology Medical University of Gdansk, Debinki 7, 80-211 Gdansk, Poland; 3Independent Laboratory of Genetic Diagnostics, Medical University of Lublin, A. Gębali 6, 20-093 Lublin, Poland

**Keywords:** acute lymphoblastic leukemia, ALL, pediatrics, *ETV6::RUNX1*

## Abstract

The most frequent chromosomal rearrangement in childhood B-cell acute lymphoblastic leukemia (B-ALL) is translocation t(12;21)(p13;q22). It results in the fusion of the *ETV6::RUNX1* gene, which is active in the regulation of multiple crucial cellular pathways. Recent studies hypothesize that many translocations are influenced by RAG-initiated deletions, as well as defects in the RAS and NRAS pathways. According to a “two-hit” model for the molecular pathogenesis of pediatric *ETV6::RUNX1*-positive B-ALL, the t(12;21) translocation requires leukemia-causing secondary mutations. Patients with *ETV6*::*RUNX1* express up to 60 different aberrations, which highlights the heterogeneity of this B-ALL subtype and is reflected in differences in patient response to treatment and chances of relapse. Most studies of secondary genetic changes have concentrated on deletions of the normal, non-rearranged *ETV6* allele. Other predominant structural changes included deletions of chromosomes 6q and 9p, loss of entire chromosomes X, 8, and 13, duplications of chromosome 4q, or trisomy of chromosomes 21 and 16, but the impact of these changes on overall survival remains unclarified. An equally genetically diverse group is the recently identified new B-ALL subtype *ETV6*::*RUNX1*-like ALL. In our review, we provide a comprehensive description of recurrent secondary mutations in pediatric B-ALL with t(12;21) to emphasize the value of investigating detailed molecular mechanisms in *ETV6::RUNX1*-positive B-ALL, both for our understanding of the etiology of the disease and for future clinical advances in patient treatment and management.

## 1. Introduction

Acute lymphoblastic leukemia (ALL) is the most common childhood cancer, accounting for more than 20% of all childhood cancers [1]. It develops by malignant transformation of immature lymphocyte precursors in the bone marrow (BM) or thymus [2]. ALL can exist in the B- (B-ALL) or T-lymphoid (T-ALL) lineage and include more than thirty distinct subtypes characterized by inherited and recurrent somatic genetic alterations that coincide with distinct gene expression profiles [3]. Currently, the survival of pediatric patients with ALL exceeds 90% [4]. Chemotherapy is administered based on stratified risk classification, which is determined by clinical factors: age and white blood cell (WBC) counts at diagnosis, genomic and cytogenetic analysis of ALL cells, and response assessment according to minimal residual disease (MRD). Karyotyping, fluorescence in situ hybridization (FISH), multiplex ligation probe-dependent amplification (MLPA), SNP microarray, next-generation sequencing (NGS) technologies such as RNA-Seq, and whole genome/whole exome sequencing (WGS/WES), are increasingly used to define more than 30 genetic subtypes. The majority of precursor B-ALL are classified in the 5th edition of the World Health Organization Classification of Hematolymphoid Tumors (WHO-HAEM5) according to aneuploidy changes, which include high hyperdiploidy (over 50 chromosomes) and hypodiploidy (less than 44 chromosomes), well-known chromosomal rearrangements (*KMT2A* rearrangements, *ETV6::RUNX1* fusion, *TCF3::PBX1* fusion, *BCR::ABL1* fusion, or *IGH::IL3* fusion), or the newly identified other genetic drivers (*BCR*::*ABL1*-like ALL, intrachromosomal amplification of chromosome 21 (iAMP21), *ETV6*::*RUNX1*-like ALL, *TCF3*::*HLF*) [5,6]. These abnormalities are, at present, routinely used in risk stratification for treatment, which has greatly contributed to outstanding improvements in treatment outcomes [3,7,8].

Certain genetic mutations that initiate the development of leukemia are acquired already in prenatal life. These mutations are the first hit, allowing certain blood cells to multiply faster than others, forming their own independent population, but this is not life-threatening. However, as we age, the number of acquired mutations increases, and if another “second hit” mutation occurs, these cells become malignant and multiply rapidly, and the result is leukemic transformation. B-ALL with *ETV6*::*RUNX1* (formerly known as *TEL::AML1*) is classified as a well-distinguished entity by the World Health Organization. Based on scientific reports, this somatic chromosomal aberration within leukemic blasts is considered a favorable prognostic indicator and is associated with improved treatment results. Nevertheless, this notion is disputed because researchers report predominantly late relapses occurring in up to 20% of patients [9]. The increasing amount of research focuses on the fact that it is not the presence of *ETV6::RUNX1* alone, but secondary rearrangements that can affect clinical implications. We focus in this study on the mechanism, significance, and treatment outcome of the most common translocation, t(12;21)(p13;q22), leading to the *ETV6::RUNX1* fusion gene that is associated with the recurrent secondary alterations.

### 1.1. ETV6

The *ETV6* gene (formerly known as *TEL*)consists of eight exons spanning 240 kb and is located on chromosome 12p13.2. It has been identified as a fusion partner in over 30 chromosomal translocation oncogenes. This gene encodes the essential hematopoietic transcriptional repressor erythroblast transformation specific (ETS) variant 6. Six isoforms of the *ETV6* are distinguished, ranging in length from 245 to 452 aminoacids [10,11]. The transcript consists of three functional domains: an N-terminal pointed dimerization domain, also termed helix loop helix (HLH), or sterile alpha motif domain, a central linker domain, and an 85-amino acid C-terminal DNA-binding domain enabling the association with the promoter region of target genes [12]. The 57-kDa ETV6 protein belongs to the ETS superfamily of transcription factors associated with numerous biological processes and pathways, such as cell growth and differentiation, hematopoiesis, angiogenesis, and megakaryocyte development. *ETV6* is highly expressed in early hematopoietic progenitor cells and is essential for hematopoiesis in the bone marrow [13,14]. At first, *ETV6* was identified as the fusion partner of the platelet-derived growth factor receptor beta gene (*PDGFRB*) in a balanced t(5;12)(q21;p13) translocation from a patient with chronic myelomonocytic leukemia. In translocations involving *MN1*, *BTL*, and *PAX5* (Figure 1), the DNA-binding domain of *ETV6* is part of the leukemogenic fusion protein, which suggests that altered expression of normal *ETV6* target genes is involved in the pathogenesis of leukemia [15,16,17]. The somatic or germline mutations in *ETV6* have been shown to contribute to malignancies such as hematologic malignancies, including myelodysplastic syndrome (MDS), acute myeloid leukemia (AML), high-risk B-ALL, early T-cell precursor (ETP) ALL, diffuse large B-cell lymphoma (DLBCL), melanoma, and colon, salivary gland, and breast cancer [18,19]. Furthermore, a relationship between heterozygous germline *ETV6* mutations to dominantly inherited thrombocytopenia and predisposition to hematologic malignancies was defined [20,21].

Melazzini et al.’s study indicated that monoallelic *ETV6* mutations result in a frequent form of inherited thrombocytopenia and a mild bleeding tendency in the affected patients, alongside the tendency towards hematologic malignancies, in particular ALL. *ETV6*-related thrombocytopenia belongs to a group of a few autosomal dominant forms of inherited thrombocytopenia without platelet macrocytosis. Therefore, screening for *ETV6* mutations in all patients with these characteristics is recommended [22].

### 1.2. RUNX1

*RUNX1* is a member of the evolutionarily conserved Runt transcription factor family. It is vital for the establishment of definitive hematopoiesis in vertebrates and is also an important regulator of the immune cells*. RUNX1* is mapped on chromosome 21q22.12 and contains nine exons spanning 150 kb. The transcript can take the form of 1 of 11 isoforms [23]. The encoded protein, which can play the role of both an activator and a repressor, is a member of the core-binding transcription factors family together with RUNX2, RUNX3, and non-DNA-binding co-factor core-binding factor beta [24,25]. *RUNX1* is essential for hematopoiesis. During hematopoietic stem cells (HSCs) differentiation, *RUNX1* expression supports cell patterning and maintaining the correct lineage. *RUNX1* is one of the most frequently mutated genes in various hematological malignancies. The alterations of *RUNX1* can result in the loss of *RUNX1* function or a dominant-negative effect. Mono-allelic *RUNX1* mutations are found in approximately 15% of T-ALL, mainly in cases with an immature phenotype and a poor prognosis [26]. In de novoAML patients, somatic mutations in *RUNX1* are detectable in 3% of pediatric and 15% of adult patients. Patients with MDS who have *RUNX1* mutations have a greater chance of developing AML. Additionally, germline mutations in *RUNX1* contribute to familial platelet disorder with a predisposition to acute myeloid leukemia (FPD/AML) occurrence. There are no statistical data regarding the frequency of FPD/AML; however, to this day, there have been over 70 families reported with this condition worldwide [27]. In addition, Latger-Cannard et al. obtained molecular characteristics of FPD/AML patients in France and found that germline *RUNX1* mutations and deletions were the most widespread alterations found in those patients [28]. Collectively, there are more than 50 different types of chromosomal translocations that affect *RUNX1* [29].

### 1.3. Detection, Mechanism, and Clinical Significance of t(12;21)

The *ETV6*::*RUNX1* fusion transcript is detected in about 17% to 25% of pediatric B-cell precursors in ALL patients, making it one of the most common non-random, recurrent translocations associated with childhood ALL [7]. Exact values vary by country: India (6%), China (18.2%), Czech Republic (22%), Greece (22.3%), Korea (23.1%), Turkey (25.5%), Italy (25.9%), Belgium (25.9%), France (27%), and Iran (34.9%). In the context of the detection of fusion genes, diverse studies have presented a significantly lower prevalence of favorable *ETV6::RUNX1* rearrangements in Mexicans (6.9–14.9%) and in Hispanics than in Caucasian or other ethnical backgrounds (25%). These studies suggested that the low occurrence of *ETV6::RUNX1* could be the result of valuable geographic and ethnic differences connected with the percentage of European and Native American genetic background in Mexicans. In the study conducted by Mata-Rocha et al., 247 samples of BM were collected from Hispanic children from Mexico City with newly diagnosed ALL. The results showed that the occurrence of the *ETV6*::*RUNX1* fusion gene transcript was at a level of 10.5% in the selected cohort. Although the reason for the variation in rates between nations is unknown, it has been suggested that they may be related to ethnic differences. Nevertheless, no significant differences in the frequency of secondary rearrangements have been observed among *ETV6::RUNX1*-positive (*ETV6::RUNX1+*) B-ALL pediatrics of different ethnical backgrounds [30,31,32,33,34,35,36,37,38,39,40]. This chromosomal translocation was first identified in 1994 by Romana et al. using the fluorescence in situ hybridization method (FISH). The t(12;21)(p13;q22) is commonly cryptic and ignored by metaphase cytogenetics, but can be detected using FISH or reverse transcriptase-polymerase chain reaction (RT-PCR) [41]. This translocation is most frequent among pediatric patients between 1 and 12 years of age, with a peak between 2 and 5 years, absent in infancy (age less than 1 year); nevertheless, between adult patients, it is rare (1–4.4%). The blast immunophenotype characteristically shows the expression of a higher intensity CD10 and HLA-DR with lower levels of the CD20, CD45, CD135, and CD34 antigens in comparison with the other genetic subtype B-ALL. In addition, *ETV6::RUNX1* cases also co-express the myeloid antigens CD13, CD33, CD65, CD27, and low or lacking expression of CD44 [42,43]. The study of Blunck et al. focused on re-evaluating the CD9 cellular expression using flow cytometry as a screening method to estimate the presence of *ETV6*::*RUNX1*. For the analysis, a total of 186 BM aspirates and/or peripheral blood samples have been collected from B-ALL pediatric patients. Subsequently, immunophenotyping of BM aspirates has been performed. The *ETV6::RUNX1* translocation has been found in 23.6% of cases. The analysis indicated that the best cut-off point for using the CD9 expression as a prediction tool for the detection of *ETV6*::*RUNX1* was 64% of the cells. Blunk et al. observed a strong association between the level of CD9 expression and the presence of *ETV6*::*RUNX1* [44].

In the course of the past decades, improvements have been made in understanding the precise role of the *ETV6::RUNX1* fusion gene product in childhood ALL. *ETV6* and *RUNX1* encode transcription factors that play important roles in hematopoiesis, and deficiency in either result in the failure of embryonic hematopoiesis. The *ETV6*::*RUNX1* transcript fuses *ETV6* exons 1 through 5 with *RUNX1* exons 2 through 8. The resulting *ETV6*::*RUNX1* fusion transcript is controlled by the *ETV6* promoter and includes the N-terminal HLH domains of *ETV6* connected to a large C-terminal part of the *RUNX1* coding sequence, containing the Runt and transactivation domains. This contrasts with *RUNX1* fusion genes found in AML, which are controlled by the *RUNX1* promoter and lack the *RUNX1* C-terminus along with the transactivation domain (Figure 2) [45].

Chimeric fusion genes are created by standard, error-prone repair of DNA double-strand breaks. Gene fusions between *ETV6* and *RUNX1* involve the noncoding introns of both genes, and the breaks are both distributed and diverse within the respective breakpoint cluster regions. The translocation t(12;21)(p13;q22) responsible for the fusion protein cause dimerization of an encoding 5′ region of the epithelium-specific ETS-like transcription factor 1 (ETS-like) and the *ETV6* gene with nearly the entire *RUNX1* gene, including its DNA-binding region and transactivation domain (Figure 3) [3]. The chromosome 12 breakpoints cluster within a single intron of the *ETV6* gene, whereas *RUNX1* breaks occur within the large and currently unsized first two introns of the *RUNX1* gene on chromosome 21. In the case of other fusion genes in leukemia, each patient’s intronic breakpoints and the following fusion sequence are unique, which provides a stable genomic marker of the derivative clone of cells [46].

The translocation results in *ETV6::RUNX1* fusion gene emergence and is associated with excellent outcomes [47,48]. Reports have shown that among patients with *ETV6::RUNX1+* ALL, the 5-year event-free survival (EFS) rate was from 80% to 97%, which was significantly higher than in other subtypes [30,48,49,50]. Nevertheless, the results of different studies indicate a high rate (20–24%) of relapse in *ETV6::RUNX1+* patients, casting doubt on the prognostic significance of this genetic mutation [51,52]. In addition, resistance to chemotherapy and relapse concern 10% of patients, which emphasizes the need for further therapy improvements [47]. *ETV6::RUNX1+* ALL is thought to arise prenatally and may be preceded by a pre-leukemic phase [53]. The controversially good prognosis of *ETV6*::*RUNX1* and long latency period in identical twins with ALL demonstrated the need for the identification of additional secondary events in disease progression. The analysis of recurrent secondary rearrangements may provide novel prognostic factors to improve current relapse treatment strategies [48].

Despite the high occurrence of t(12;21) in B-ALL, the translocation was shown to be insufficient to induce leukemic transformation on its own [13]. The *ETV6::RUNX1* fusion protein induces a silent preleukemic clone. The *ETV6::RUNX1*+ HSCs are able to self-renew, and they contribute to hematopoiesis; however, they differ from typical HSCs [51]. The translocation is usually acquired in utero in very early progenitor cells, prior to T- or B-cell receptor gene rearrangements [54]. As mentioned above, additional genetic hits are needed for the transition to leukemia. The further alterations are acquired independently with no preferential order, and they may include loss of *ETV6* or alterations directed to genes regulating regular B-cell differentiation [52]. These genetic modifications can be obtained after years of latency and lead to ALL development (Figure 4) [29].

According to research on monozygotic twins with concordant leukemia, chromosomal translocations, which are typical for pediatric leukemia, often occur prenatally. Based on this hypothesis, preleukemic clones, functional fusion genes, and chromosome translocations should be far more frequent in healthy newborns’ blood than the general risk of developing the associated leukemia. Following the analysis of 253 healthy newborns’ cord blood, Zuna et al. state that 2% of newborns carried *ETV6::RUNX1* fusion. In addition, liver and spleen tissue samples from aborted fetuses were tested for the presence of the *ETV6::RUNX1* mutation, and the spleen tissue was found to harbor the mutation [55]. To analyze the prevalence of *ETV6::RUNX1+* preleukemic cells in newborns, Fueller et al. developed a DNA-based method of genomic inverse PCR for the exploration of ligated breakpoints (GIPFEL) that enables the sensitive recognition of recurrent chromosomal translocations. A total of 50 out of 1000 samples were detected with *ETV6::RUNX1* gene fusion, establishing that more than 5% of newborns harbor the *ETV6::RUNX1* gene fusion. Translocation has been shown to be present frequently during regular fetal development and is 100 times more common than developing leukemia (1 in 10,000 children) [56]. Mori et al. confirmed previous reports that the frequency of *ETV6::RUNX1* fusion genes commonly found in leukemia in cord blood is 100 times higher than the chance of developing the corresponding leukemia in the future. *ETV6::RUNX1+* cells were found in more than 1% of cord blood from newborns. This indicates that *ETV6::RUNX1* has a very low oncogenic potential and prophylactic screening of newborns carrying *ETV6::RUNX1* is limited, considering that 99% of newborns will never develop the disease. These findings emphasize the significance of spontaneous or environmentally induced secondary hits in *ETV6::RUNX1+* ALL [57].

## 2. Secondary Genetic Rearrangements

Numerous studies demonstrated that *ETV6*::*RUNX1* rearranged B-ALL patients harbored a high number of copy number alterations (CNAs).

A study using the array comparative genomic hybridization carried out on eleven pediatric *ETV6::RUNX1+* ALL patients by Zakaria et al. showed a significant amount of additional chromosomal aberrations: 119 losses, 36 gains. The number of aberrations per patient ranged from 3 to 58, with a mean of 14 aberrations per patient [58].

Zhang et al. sequenced 120 candidate genes in 187 patients with high-risk B-ALL. A total of eight *ETV6* sequence mutations were detected in six patients. When *ETV6* sequence mutations were combined with somatic copy number deletions, 12% had loss-of-function mutations in this gene, with 18 of 23 being monoallelic and 5 of 23 biallelic [59].

Rare anomalies in childhood ALL are near-tetraploidy (82–94 chromosomes), which was observed in *ETV6::RUNX1*+ ALL by Attarbaschi et al. The authors did not explain whether tetraploidization also is such a comparatively early evolutionary step that merely promotes the malignant potential of *ETV6::RUNX1+* blast cells, or whether it is actually rather a late event or even the last trigger that sets off the clinical disease manifestation [60].

Borst et al. used SNP arrays to investigate secondary rearrangements found in pediatric Scandinavian patients enrolled in the NOPHO treatment protocol. After analyzing a total of 62 patients, 129 deletions were detected, and 59% of them ranged over 1Mbp. Interestingly, of a total number of 45 gains, 44 of them were above 1 Mbp. In only 13 patients, no CNAs were detected [61].

There is an abundance of research articles regarding secondary rearrangements in B-ALL; however, there is very little research regarding *ETV6::RUNX1*+ ALL specifically. Certain research contains percentage data on the prevalence of secondary mutations in *ETV6::RUNX1*+ ALL; however, their clinical relevance has not been established yet. In B-ALL, there are mutations whose presence allows for the stratification of patients upon the beginning of the treatment into low- or high-risk groups. Therefore, based on the existing knowledge, it can be assumed that those secondary mutations that have been associated with another B-ALL subgroup could be present as “second hit” mutations in the *ETV6::RUNX* group and have similar diagnostic and clinical implications. The described recurrent secondary genetic deletions in *ETV6:RUNX1+* B-ALL in order from most to least frequent are deletions of 12p, 6q, 9p, 5q, 3p, 3q, 14q, 7p, 7q, 4q, 19q, 11q, 11p, 1q, 13q, and 8p, and the most common duplications are 4q, Xq [59,62,63,64,65]. The summary of these alterations related to *ETV6*::*RUNX1*+ B-ALL is presented in Table 1.

### 2.1. Chromosome 12

The most common secondary alteration detected in approximately 70% of ALL patients with *ETV6::RUNX1* includes the loss of the normal or non-rearranged *ETV6* homolog. *ETV6* can function as a tumor suppressor and dimerize with *ETV6::RUNX1* to reduce its transforming activity [59].

The majority of studies have reported deletion region 12p in patients with *ETV6::RUNX1* [71]. This region is abundant in genes that are vital for B-cell development, such as *CDKN1B* (Cyclin Dependent Kinase Inhibitor 1B) and *BCL2L14* (*BCL2-Like 14)*. *CDKN1B* is a negative regulator of cell-cycle progression from the G1 to S phases. Deletions of *CDKN1B* have been observed in pediatric ALL as well as murine models of this disease [72]. *BCL2L14* is a pro-apoptotic gene, therefore acting as a tumor suppressor [73,74].

The expression of *BTG1* (B-cell translocation gene Anti-Proliferation Factor 1) is located on chromosome 12q21.33. The genotoxic and cellular stress signaling pathways, which both affect the activity of transcriptional co-factors and regulate posttranscriptional mRNA stability, are thought to be mediated by *BTG1* and *BTG2*, respectively [75]. Research conducted on pediatric patients by Waanders et al. showed that deletion of *BTG1* was present in 27 out of 142 (19%) *ETV6::RUNX1+* cases [76].

Interestingly, *ETV6* deletions were detected less frequently among patients with clonal RAS pathway mutation [77]. Nevertheless, there has been a great deal of differing research revealing the scarce, yet definite, presence of *RAS* mutations in patients with *ETV6::RUNX1+* ALL. Nishii et al. analyzed different variants of *ETV6* and assessed their transcriptional profile. Germline *ETV6* variants were associated with hyperdiploid B-ALL, whereas somatic *ETV6* mutations were found in *ETV6::RUNX1+* ALL. In cases presenting “damaging *ETV6* variants”, the aberrant transcription was driven by hyperdiploidy as well as gene aberrations such as RAS and NRAS pathway mutations. Without *NRAS* mutation, the cells harboring somatic *ETV6* mutation did not undergo oncogenic transformation [78].

### 2.2. Chromosome 6

*CCND3* gene is located on 6p21.1, and mutation of *CCND3* results in cell-cycle deregulation. In the study by Ueno et al., with the use of panel sequencing, *ETV6::RUNX1+* ALL was shown to be correlated with *CCND3* mutation. The alteration was identified in 13 cases of the study, 8 of which (62%) were *ETV6::RUNX1+* [64]. Van Delft et al. investigated the clonal origins of relapse by comparing the profiles of genome-wide CNA presented in 21 patients with those in matched relapse. There were identified, in total, 159 copy number alterations at presentation and 231 at relapse. Deletions of *CDKN2A/B* or *CCNC* (6q16.2-3), or both, increased from 38% at presentation to 76% in relapse, suggesting that cell-cycle deregulation contributed to the emergence of relapse [79]. *RUNX1* gene is not a potent transcriptional regulator by itself. Instead, interactions with other transcriptional regulators such as transcription factors (*FOX3*, *GATA2*, *SCL1*, *ERG*), co-activators, or co-repressors, as well as post-translational changes that impact influence *RUNX1* transcriptional activity [80].

### 2.3. Chromosome 21

*RUNX1* gene has been characterized as a molecular switch that regulates the balance between proliferation and differentiation during development. Pais et al. conducted a large cohort study of 515 patients, in which 77 cases showed *ETV6::RUNX1* fusion. In the *ETV6::RUNX1+* group, 70% had additional *ETV6::RUNX1* homolog changes. The changes that affect the *RUNX1* gene are the presence of additional *RUNX1* signals and the loss of residual *RUNX1* [65].

SPI-B (encoded by *SPIB*) is an important transcriptional activator of B-cell development and differentiation. SPIB mRNA transcript levels are low in *ETV6::RUNX1*+ ALL relative to other leukemia subtypes, as the *SPIB* gene is directly activated by *RUNX1* during B-cell development. Xu et al. hypothesized that SPIB is directly transcriptionally repressed by *ETV6::RUNX1*; therefore, using chromatin immunoprecipitation, they identified a regulatory region in the first intron of SPIB that interacts with *ETV6::RUNX1* [81]. Additionally, the research conducted by Pang et al. proves that SPI-B expression is significantly reduced in *ETV6::RUNX1*+ ALL. Therefore, p21 and SPI-B are important targets of *ETV6::RUNX1* in the regulation of B-cell gene expression in leukemogenesis [82].

To evaluate the impact of *RUNX1* and *ETV6::RUNX1* on RAG1 expression, Jakobczyk et al. used human B-cell precursor Nalm6 cells. In their study, enforced expression of *ETV6::RUNX1* or *RUNX1* in Nalm6 cells intensified the levels of endogenous RAG1 transcript and protein. Those results validated the direct or indirect upregulation of RAG1 expression by both *ETV6::RUNX1* and *RUNX1* [83].

### 2.4. Chromosome 9

Up to 25% of *ETV::RUNX1+* children have the 9p deletion, which frequently impacts cyclin-dependent kinase inhibitor 2A or 2B *(CDKN2A/CDKN2B)* and the B-cell differentiation regulator *PAX5* [84]. *PAX5* is a gene located in 9p13. The absence of *PAX5* activity causes developmental arrest and the lack of typical B-lymphoid lineage commitment. In mature B-cells, the inactivation of *PAX5* leads to the loss of their identity. In a physiological complex containing *IKZF1* and *RUNX1*, *PAX5* can be discovered. *PAX5* works as a metabolic gatekeeper with *IKZF1* to regulate the distribution of glucose and energy. This limitation is lifted by heterozygous *PAX5* deletion, which also boosts ATP levels and glucose absorption [85]. Agarwal et al. observed CNAs in the population of Indian pediatric patients. In a cohort of 91 B-ALL patients, 30.7% showed significant cytogenetic alterations. Four out of nine *ETV6::RUNX1+* patients had *ETV6* deletion, one patient had *BTG1* deletion, and one patient had *PAX5* deletion. It was stated that 12 months after induction chemotherapy, patients who were found with deletions in *PAX5* and *ETV6* had better survival outcomes as compared to those without gene alterations [86].

### 2.5. Chromosome 5

*NR3C1* (Nuclear Receptor Subfamily 3 Group C Member 1) is a protein-coding gene that encodes a glucocorticoid receptor. Kuster et al. examined single nucleotide polymorphism arrays for DNA CNAs in 18 matched diagnoses and relapse leukemias to comprehend the molecular background of relapse mechanisms. Only 21.4% of patients had totally new mutations at relapse, whereas the additional 78.6% had a common prototype and later acquired distinctive CNA. Researchers detected recurring, generally nonoverlapping deletions related to apoptosis caused by glucocorticoids. It is targeted by the Bcl2 modifying factor (BMF) and the glucocorticoid receptor NR3C1. In the study by Kuster et al., 24 patients with relapse and 72 without relapse of *ETV6::RUNX1+* were tested by FISH, and the results showed that BMF deletions were always present at diagnosis, while *NR3C1* and mismatch repair aberrations predominated in relapse. The above-mentioned genes’ alteration was connected to leukemias’ treatment failures [87]. In Grausenburger et al. study, deletions involving the glucocorticoid receptor gene *NR3C1* were particularly associated with poor response to induction therapy [88].

### 2.6. Chromosome 7

*IKZF1* (Ikaros zinc finger transcription factor1), located on 7p12.2, regulates hematopoiesis and is essential for lymphoid differentiation [89]. Mullighan et al. proved that deletion or mutation of *IKZF1* is related to poor treatment outcomes and a high risk of relapse among pediatric ALL patients [90]. Enshaei et al. investigated 368 *ETV6::RUNX1+* and stated that six *IKZF1*-deleted patients have remained in complete remission for over 9.5 years [73].

*IGF2BP* (Insulin-like growth factor 2 mRNA-binding protein), located on 7p15.3, is a family of oncofetal RNA-binding protein genes consisting of *IGF2B1*, *IGF2B2*,and *IGF2B3* [91]. In recent years, it has been proposed as a novel marker for *ETV6::RUNX1* by Sharma et al., who observed that the expression of both *ETV6::RUNX1* and *IGF2BP1* correlated significantly with the blast percentage. Interestingly, the relative expression of *ETV6::RUNX1* was much lower than *IGF2BP1* levels contributing to the higher sensitivity of *IGF2BP1* expression [92]. Research conducted by Stoskus et al. showed that among patients with *ETV6::RUNX1* fusion, *IGF2BP1* and *IGF2BP3* were expressed at a higher level than *IGF2BP2* [93]. Similar results were obtained by Mäkinen et al., who detected high levels of I*GF2BP3* in patients with *ETV6::RUNX1* [94].

### 2.7. Chromosome 11

*KMT2A* (lysine methyltransferase 2A) located in the 11q23 region gene encodes a transcriptional co-activator, which is crucial for controlling gene expression throughout hematopoiesis and early development [95]. Pais et al. reported a unique finding, which is the co-existence of *ETV6*::*RUNX1* and *KMT2A* (previously known as *MLL*) aberrations. Translocation, monoallelic deletion, or rearrangement of *KMT2A* were identified in 9% of patients. Results suggested that *KMT2A* aberrations in the *ETV6::RUNX1+* group had no prognostic significance [65].

Attarbaschi et al. classified *KMT2A* as a non-random and highly specific secondary aberration in *ETV6::RUNX1+* ALL, rather than a disease-relevant primary abnormality. Their results provided evidence that ALL with *KMT2A* abnormalities represents a highly distinct and separate subgroup of ALL. Based thereon, they suggested that in future clinical ALL studies, cases with a cytogenetically evident *KMT2A* should not be evaluated together with *KMT2A* translocations anymore, but should be screened for the *ETV6::RUNX1* fusion gene right away [60].

Papaemmanuil et al. performed a genomic analysis on 57 patients with *ETV6::RUNX1+* ALL, which showed that RAG-initiated deletions, characterized by recombination signal sequence motifs near breakpoint junctions, appear as the predominant mutational process. A common target of deletions was the RAG locus on 11p12. What is more, Papaemmanuil et al. state that the major secondary events leading to leukemic transformation in *ETV6::RUNX1+* ALL are often caused by genomic rearrangements mediated by aberrant RAG recombinase activity, and rarely by point mutations [96].

## 3. *ETV6::RUNX1+* Prognosis and Treatment

The prognostic significance of the *ETV6*::*RUNX1* fusion is still analyzed by scientists, as the results of the prognosis studies of this subgroup are ambiguous. Already in the 1990s, there were discrepancies regarding the prognosis of patients with *ETV6::RUNX1* in B-ALL. The reports noted that as many as 25% of children on BFM protocols who suffer from relapses were *ETV6::RUNX1+* [97].

Loh et al. conducted a prospective study to determine the incidence and outcomes of children with *ETV6::RUNX1+* ALL. Patients were risk stratified primarily by current National Cancer Institute (NCI)–Rome risk criteria. According to their results, the 5-year OS rate was 97% among *ETV6::RUNX1+* patients compared with 89% among *ETV6::RUNX1−* patients. The analysis concludes that patients with *ETV6*::*RUNX1*+ usually have excellent outcomes. However, factors such as age at diagnosis and presented leukocyte count should be considered during treatment [51].

Ampatzidou et al. studied, in a pediatric cohort of 119 B-ALLs, the relation between the *ETV6*::*RUNX1* aberration and the co-existing subclones with clinical and biological features (sex, WBC, organ infiltration, karyotype, additional molecular aberrations), early response to treatment (MRD) and long-term outcome. In the study group, 22.7% of patients were *ETV6*::*RUNX1*+ and 70.4% harbored additional genetic abnormalities, while 33.3% presented clonal heterogeneity. A common feature of all recurrences was subclonal heterogeneity, FCM-MRD d15-positivity, and additional del(9p21). There were no altered recurrences among *ETV6::RUNX1+* patients. In this study, the presence of clonal heterogeneity and impaired FCM-MRD clearance among *ETV6::RUNX1+* patients influenced prognosis, but the author suggested that a longer follow-up was needed [62].

Qiu et al. conducted a multicenter, retrospective study on 2530 children to analyze the outcomes of *ETV6*::*RUNX1*+ pediatric B-ALL in South China, with the aim of identifying significant prognostic variables. *ETV6*::*RUNX1*+ translocation was present in 18.2% of cases. Qiu et al.’s study results demonstrate that chemotherapy protocol, WBC, age, and mixed-lineage leukemia gene rearrangement (MLL-r) status are all independent, significant predictors of the outcome among childhood B-ALL. Notably, this study mentioned that a large percentage of *ETV6::RUNX1+* patients had moderate anemia and thrombocytopenia at the diagnosis onset [30].

The group of Piette et al. evaluated the long-term prognostic and predictive value of *ETV6::RUNX1* in *BCR::ABL1* negative de novo B-ALL children. Particular attention was given to the effects of the randomized treatments in the *ETV6::RUNX1+* subgroup as compared with those observed in the high hyperdiploid and other analyzed B-ALL subgroups, in order to reveal specific drug response profiles related to distinct oncogenic processes. The 10-year EFS rate was 73.1–82.7%, but it was significantly higher in the *ETV6::RUNX1+* than in the *ETV6::RUNX1−* group. *ETV6::RUNX1+* patients also had a significantly higher 10-year OS rate exceeding 95.3%. It was remarkable that in the *ETV6::RUNX1+* group, EFS events mostly occurred after the end of the maintenance therapy, with very few events before, and virtually no event after 6 years from diagnosis [37]. Pui et al. conducted a study where they incorporated more stringent risk classification, early intensification of intrathecal chemotherapy, reinduction treatment, and the use of prednisone instead of the commonly used dexamethasone in post-remission therapy. They found that sex, age, race, immunophenotype, CNS status, and the t(1;19)/E2A-PBX1 or *ETV6*::*RUNX1* lacked prognostic significance [98].

One of the first examinations of the chemosensitivity of primary leukemic cells from children with newly diagnosed t(12;21)-positive B-ALL was conducted by Narla et al., whose results provided unprecedented evidence that *ETV6::RUNX1+* ALL cells are more sensitive to dexamethasone and vincristine than ALL cells without *ETV6::RUNX1* [99].

Ramakers-van Woerden et al. investigated the relationship between the presence of *ETV6::RUNX1* and in vitro drug resistance in pediatric B-ALL using cells obtained from the BM or peripheral blood of 192 children with newly diagnosed ALL. The *ETV6::RUNX1+* patients were slightly, but significantly, more resistant to vincristine and cytarabine, and comparatively more sensitive to l-asparaginase (L-ASP). These results imply that *ETV6::RUNX1+* patients could benefit from therapy regimens with significant use of L-ASP [100].

To estimate whether outcomes for *ETV6*::*RUNX1*+ ALL are improved by contemporary risk-directed therapy, Bhojwani et al. studied clinical features, response, and adverse events of 168 children with newly diagnosed *ETV6*::*RUNX1*+ ALL in St. Jude Total Therapy studies XIIIA, XIIIB, and XV. Results were compared with 494 *ETV6*::*RUNX1*-negative (*ETV6*::*RUNX1*-) B-ALL patients. The incidence of *ETV6::RUNX1* was associated with a patient age between 1 and 9 years, pre-treatment classification as low risk, and lower levels of MRD at 19 days of therapy. EFS or OS did not differ between patients with or without *ETV6*::*RUNX1* in Total XIIIA or XIIIB. By contrast, in Total XV, patients with *ETV6*::*RUNX1* had significantly better results than those without *ETV6*::*RUNX1* fusion EFS (97% vs. 88%) and OS (99% vs. 94%). The MRD-guided treatment schema that included intensive asparaginase and high-dose methotrexate in the Total XV study produced significantly better outcomes than previous regimens, and demonstrated that nearly all children with *ETV6*::*RUNX1+* ALL can be cured with risk-directed therapy including intensive asparaginase, vincristine, dexamethasone, and high-dose methotrexate [50].

A study conducted by Wang et al. proved a good prognosis for the *ETV6::RUNX1+* ALL subgroup based on the clinical characteristics and treatment outcomes of 77 pediatric patients with *ETV6::RUNX1+* ALL. The 5-year EFS and the disease-free survival (DFS) were reported to be 90% and 96%, respectively [49].

In general, maintenance therapy lasts over a year and consists of daily mercaptopurine and weekly methotrexate intake, and its role is to eradicate residual leukemic cells [79]. Shortening of maintenance therapy (terminated at 1 year after initiation of treatment) in ALL patients is thought to result in a high relapse rate [101]. However, studies indicate that some genetic subgroups, including *ETV6::RUNX1*, were associated with excellent DFS despite completing maintenance therapy at one year after diagnosis [102].

Kato et al. conducted a study to identify groups of patients who can be cured even with a shortened duration of maintenance therapy. After the comparison of genetic analysis results with clinical data, Kato suggested that short-duration therapy can cure more than 50% of pediatric ALL, especially in females and patients with *TCF3::PBX1* or *ETV6::RUNX1* mutations [103].

The translocation (12;21) had a strong impact on OS after relapse. Gandemer et al. conducted a long-term, follow-up retrospective study based on patients included in the French group for the childhood ALL 93 trial (FRALLE 93) protocol to evaluate the outcomes of patients with relapsed *ETV6::RUNX1*+ ALL. Among the 713 children with ALL tested for *ETV6::RUNX1*, 191 patients were *ETV6::RUNX1*+, of which 19.4% had a relapse. The three-year survival rate for *ETV6::RUNX1*+ patients totaled 64.7%, compared to 46.5% for *ETV6::RUNX1*-. *ETV6::RUNX1*+ ALL male patients had a longer remission-free period compared to patients with other B-ALL. However, in *ETV6::RUNX1*+ male patients, a significant increase was noticed in testicular cancer relapses. The authors point out that among patients with *ETV6::RUNX1* mutation, more attention should be focused on long-term follow up to relapses in gonads or usage of chemotherapeutics that do not cross the testicular barrier [38].

## 4. *ETV6::RUNX1+* Future Treatment Perspective

Despite significant improvements in the treatment of children with ALL, it is emphasized that the intensity of conventional chemotherapy has reached the limit of tolerance. Therefore, cytostatic dosing can no longer be increased with the hope of achieving better results. This reinforces the need to search for a new solution to toxic chemotherapy, such as molecular and immune therapies. Pui suggested that among different ALL subgroups, children with *ETV6::RUNX1* notably appear to be suitable candidates for reduced standard chemotherapy treatment [98]. In the Associazione Italiana di Ematologia e Oncologia Pediatrica (AIEOP) and BFM ALL 2000 protocol, patients from 1to 17 years of age with standard-risk B-ALL were randomized to receive standard treatment or reduced delayed intensification. With the exception of individuals with *ETV6::RUNX1+* ALL or patients of one to six years of age, who responded equally well toall treatment regimens, this modification influenced total eight-year DFS, which varied from 89.2% to 92.3% between the experimental reduced-intensity group and standard treatment group [104].

It has been established that cytotoxic chemotherapeutics, frequently applied in the treatment of leukemia, act by promoting apoptosis in sensitive target cells. Faulty apoptosis may contribute to the development of leukemia resistance to chemotherapy. Several positive and negative regulators of apoptosis have been identified, including the pro-apoptotic receptorCD95/Fas and the anti-apoptotic protein BCL2. Narla et al. conducted an in vitro study on the chemosensitivity of primary leukemic cells from newly diagnosed *ETV6::RUNX1+* ALL patients vs. *ETV6::RUNX1-* ALL cells. Their results indicated that *ETV6::RUNX1+* ALL cells expressed higher levels of the pro-apoptotic protein Fas and lower levels of the anti-apoptotic protein Bcl2 than *ETV6::RUNX1−* ALL cells. Moreover, the *ETV6::RUNX1+* ALL cells were more sensitive to the apoptosis-inducing effects of serum deprivation, dexamethasone, and vincristine than *ETV6::RUNX1−* ALL cells [99].

Target therapy is not currently available for children with *ETV6::RUNX1+* ALL. To identify the molecular mechanisms underlying *ETV6::RUNX1+* ALL, Polak et al. performed gene expression profiling in healthy hematopoietic progenitors ectopically expressing *ETV6::RUNX1*. In that process, they revealed a transcriptional network driven by *ETV6::RUNX1* that induces proliferation, survival, and cellular homeostasis. Moreover, Vps34, an important regulator of autophagy, was found to be induced by *ETV6::RUNX1* and upregulated in *ETV6::RUNX1+* cells. Inhibition of Vps34 in *ETV6::RUNX1*+ cell lines severely reduced proliferation and survival. Hydroxychloroquine was applied to inhibit autophagy, which reduced cell viability in both *ETV6::RUNX1*+ cell lines and primary ALL samples. In addition, it selectively sensitized primary *ETV6::RUNX1*+ ALL samples to L-asparaginase [53].

Serafin et al. aimed to assess the role of spleen tyrosine kinase (SYK) in cells with *ETV6::RUNX1* translocation from B-ALL patients. In order to sensitize the resistant primary cells to conventional medications, Serafin treated cells from patients both at the time of diagnosis and after relapse with the combination of entospletinib and chemotherapeutics. Their results suggested thatentospletinib is able to induce cell death and enhance the efficacy of conventional chemotherapeutics. However, more extensive studies considering other SYKs are required [105].

Akbari et al. proposed an application of homologous recombination (HR) using an in vivo model of *ETV6::RUNX1* as a way of targeted therapy in *ETV6::RUNX1+* ALL. The results showed that the expression of *ETV6::RUNX1* was significantly decreased in the HR-edited cells. Moreover, the edited cells had decreased viability in comparison with non-edited cells. Hopefully, in the future, modification by the HR technique may have the effect of reducing the leukemic effect of the fusion associated with the *ETV6* and *RUNX1* genes; however, it is still too early to introduce this method into clinical practice [106].

## 5. *ETV6::RUNX1*-like

In recent years, multiple groups from the USA, Europe, Japan, and China have generated or used B-ALL RNA-seq data to identify new targets of recurring rearrangement (e.g., *DUX4*, *MEF2D*, and *ZNF384*) associated with distinct gene expression profiles and the presence of cases with alterations that phenocopy additional canonical B-ALL drivers, e.g., *ETV6::RUNX1*-like ALL [107,108]. *ETV6::RUNX1*-like occurs in about 2–3% of children and less than 1% of adult patients with B-ALL. *ETV6::RUNX1*-like is a subtype of B-ALL defined by the *ETV6::RUNX1-*specific gene expression profile harboring concurrent *ETV6* and *IKZF1* lesions, but no exact *ETV6::RUNX1* gene fusion. Both *IKZF1* and *RUNX1* encode transcription factors crucial for B-cell maturation, and there is speculation that the loss of *IKZF1* may replace the altered function of *RUNX1* in the *ETV6::RUNX1* fusion protein. Consistent with this, Lilljebjörn et al. note that *IKZF1* deletions are rare in *ETV6::RUNX1*+ patients and occur in approximately 3% of them. Among the changes in *ETV6::RUNX1*-like ALL patients, the following fusions are included: *ETV6* and *PMEL*; *IKZF1* and *CDK2; ETV6* and *BORCS5*; *SETD5* and *IKZF1*; *ETV6* and *NID1*; *ETV6* and *CREBBP*; *ETV6* and *BCL2L14*; and *ETV6* and *MSH6*. In addition, deletions of the second *ETV6* allele and 7p were detected [107].

At the early stages of B-cell development, the precursors undergo immunoglobulin gene rearrangement, which is necessary for BCR formation on mature B-cells. Studies on mice have shown that recombination-activating genes (RAGs) are essential for this process. Chen et al. analyzed 1582 ALL patients, using both microarray and RNAseq data sets, to determine which subtypes express RAG1 or RAG2 and if any genes are co-expressed whose presence could identify new subtypes. Chen et al. analyzed the expression of RAG1 and RAG2 genes. Both genes were detected in the *ETV6::RUNX1* subtype. In particular, RAG1 expression was consistently higher in the *ETV6::RUNX1* compared to all other genetic subtypes, except the B-other ALL group. In a genomic screen for genes co-expressed with RAG1 and RAG2, none appeared consistently with RAG2, but an expression set of 31 genes was identified with RAG1. Using RAG1 along with the co-expression genes as an identifier distinguishes *ETV6::RUNX1* from all other B-ALL subtypes, except for five cases. *ETV6* target genes *CLIC5*, *WBP1L*, *ANGPTL2*, and *BIRC7* were expressed at very high levels in both *ETV6::RUNX1* and *ETV6::RUNX1*-like samples, with a consistent pattern and levels. The results of the Chen et al. study suggest that the *RAG1* signature identifies a new *ETV6::RUNX1-*like subtype, since there are no definitive genetic markers that identify this new subtype [109].

A recent study by Zaliova et al. aimed to answer the question of whether *ETV6::RUNX1-*like can be distinguished by a specific immunophenotype. During diagnostic immunophenotyping of 573 pediatric B-ALL, researchers, among the B-other ALL patient group, identified eight cases with characteristic immunophenotypes for *ETV6::RUNX1+* patients. The results suggest that the expression pattern of two surface proteins, CD27-positive and CD44-low-negative, distinguish *ETV6::RUNX1*-like ALL from other B-ALL. In addition, in genetic studies, the two most common alterations associated with *ETV6* loss were various *IKZF1* mutations and *ARPP21* deletions [110].

## 6. Conclusions

The most prevalent genetic aberration in pediatric B-ALL is the t(12;21)(p13;q22) translocation. Accurate characterization of the disease genotype remains a priority, as it forms the basis for diagnosis, prognosis, risk assessment, and choice of therapy. Studies indicate that depending on the nature of the second hit, *ETV6::RUNX1* ALL can be divided into subtypes based on the gene expression profile. Integrating the knowledge about *ETV6::RUNX1* fusion and secondary molecular rearrangements into a treatment approach could reduce patients’ chance of relapse and improve outcomes.

## Figures and Tables

**Figure 1 cells-12-00357-f001:**
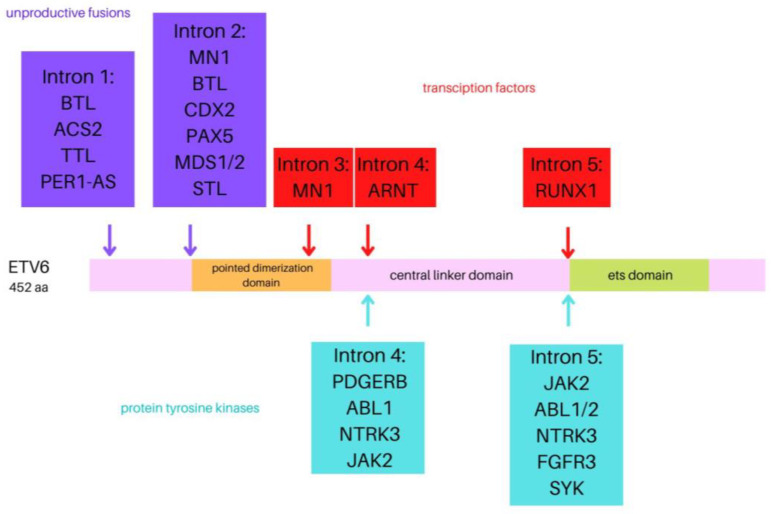
Visual representation of the ETV6 protein with the position of the breakpoints of the various fusion partner genes.

**Figure 2 cells-12-00357-f002:**
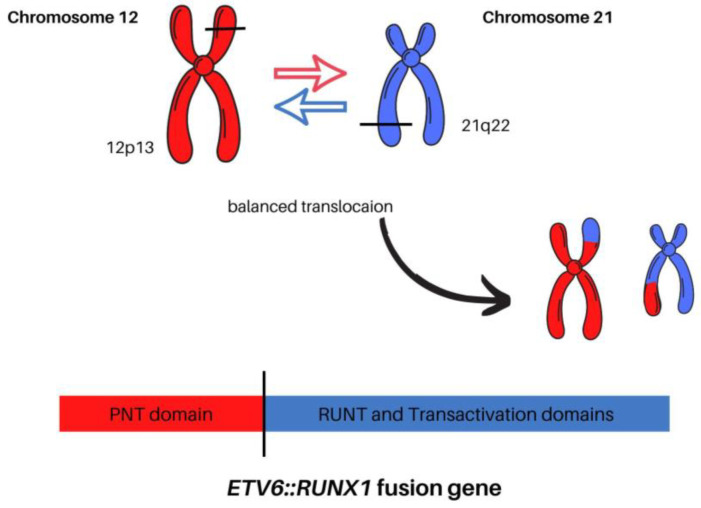
The comparison between normal chromosomes 12 and 21 versus the t(12;21)(p13;q22) translocation.

**Figure 3 cells-12-00357-f003:**
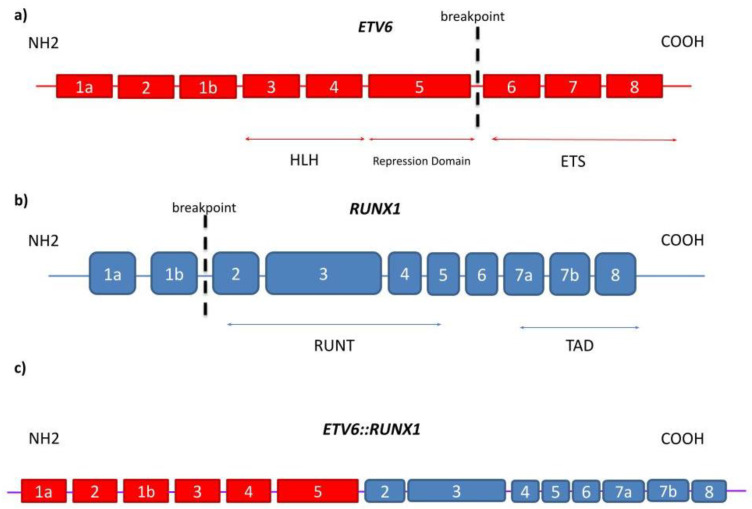
(**a**) On *ETV6* gene, which consists of nine exons, the translocation occurs between exon 5 and exon 6; (**b**) On *RUNX1* gene, the breakpoint is located between exon 1b and exon 2 out of 10 exons in total; (**c**) the fusion results in formation of *ETV6::RUNX1* translocation which is transcribed into ETV6::RUNX1 fusion protein.

**Figure 4 cells-12-00357-f004:**
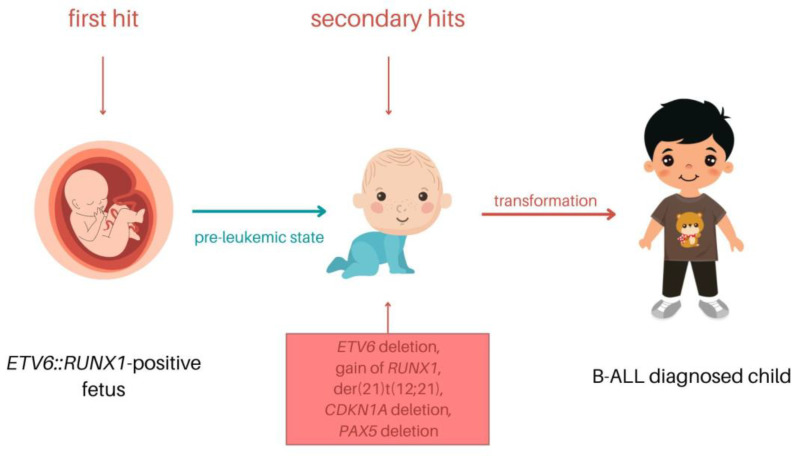
The model of the pediatric *ETV6::RUNX1* B-ALL process.

**Table 1 cells-12-00357-t001:** Recurrent secondary genetic changes found in childhood *ETV6::RUNX1* B-ALL.

Chromosome Number	Observed Abnormality	Possibly Covered Genes	Frequency of Occurrence	Reference
12	deletion of 12p	*ETV6*, *CDKN1B*, *BCL2L14*, *BTG1*, *KRAS*	12–39%	[58,66,67]
6	deletion of 6q	*AIM1*, *PRDM1*, *FOXO3*, *CCNC3*, *FYN*, *CDKN1A*	13–33%	[66,68]
21	gain of normal chromosome 21 gain of the der(21)t(12;21)(p13;q22)	*RUNX1*	25%	[61,67]
9	deletion of 9p	*CDKN2A/B*, *PAX5*, *MTAP*, *JAK2*, *P14ARF*, *P16IKN4a/ARF*	10–25%	[58,66]
5	deletion of 5q	*NR3C1*, *EBF1*	23%	[69]
3	deletion of 3p deletion of 3q	*LIMDI*, *ARPP-21*, *ULK4*, *FHIT*, *CD200*, *BTLA*, *TBL1XR1*	3–21%	[66,67]
14	deletion of 14q	*DPF3*	3–21%	[58]
7	deletion of 7q deletion of 7p	IKZF1, IGF2BP	3–18%	[2,58]
4	duplication of 4q deletion of 4q	*NR3C2*, *YIPF7*, *ARHGAP10*	6–17%	[66]
19	deletion of 19q	*CEBPA*, *UHRF1*, *GRLF1*, *NPAS1*, *TMEM160*	6–13%	[67]
X	monosomy X in females/ gain of X in males gain of Xp duplication of Xq	*SPANXB*, *HMGB3*, *FAM50A*, *HTATSF1*	4–11%	[66]
11	deletion of 11q deletion of 11p	*CD44*, *RAG1/2*, *BACL2*, *GNG3*, *HNRPUL2*, *TTC9C*, *ATM*, *KMT2A*, *HRAS*	10%	[67,68]
1	deletion of 1q	*TROVE2*, *GLRX2*, *CDC73*, *B3GALT2*, *PDE4B*, *NRAS*,	10%	[69]
15	deletion of 15q	*LTK*, *MIRN626*	10%	[70]
13	deletion of 13q/monosomy	*BTG1*, *RB1*, *SERP2*, *DLEU1/2/7*, *STBP4*, *TRIM3*, *KCNRG*, *MIRN16-1*, *MIRN15A*	5–10%	[58]
8	deletion of 8p	*CTSB*, *LOXL2*, *NKX3-1*, *WHSC1L1*,*FGFR1*, *IDO1*, *IDO2*, *KAT6A*	6–8%	[58,68]

## Data Availability

Not applicable.

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
