# Peer review of "The Landscape of Secondary Genetic Rearrangements in Pediatric Patients with B-Cell Acute Lymphoblastic Leukemia with t(12;21)"

_cells, 2023, doi:10.3390/cells12030357_

Round 1
Reviewer 1 Report
The manuscript is ambitious and reading the abstract it is captivating and interesting. Despite this, however, some changes are necessary for the text to be scientifically effective.
The major points are the followings:
1) The main text is unbalanced considering both the title and the abstract: indeed, the treatment and prognosis paragraph is too long and not completely relevant, in fact in the abstract the prognosis is just mentioned and in fact it is not the main topic of the review. Thus, I suggest to shorten the “ETV6::RUNX1+ treatment, prognosis” paragraph. Minor point: there is no numbering in that paragraph, probably it is the number 3, please check it. Page 12, sentence “The prognostic significance… disputable”: I disagree with the term disputable and I would consider the overall prognosis studies of this subgroup.
2) The paragraph n°2, entitled “Secondary genetic rearrangements”, is very comprehensive and it takes into account several publications. However, as it is written, it is not very effective, as it is too descriptive and seems a list of secondary events, without meaning or as single events. Instead, the authors should rewrite it trying to interpret the meaning of these second hits. The authors can propose their own interpretation, for example according to the type or function of altered genes, or even considering whether they act on the same pathways as ETV6 or RUNX1.
3) Along the same lines, the "two-hit" model is mentioned in the abstract which is absolutely pertinent to the subject of the review, but in the text, there is only a hint of it, in the text and with figure 4. I would deep this concept.
4) Considering the mentioned references, in my opinion relevant publication are missing or more recent ones than the already cited ones:
- in addition to ref. 5: “International Consensus Classification of acute lymphoblastic leukemia/lymphoma”. Amy S. Dufeld et al. 2022, regarding WHO classification.
- In addition to ref. 6: “Has Ph-like ALL Superseded Ph+ ALL as the Least Favorable Subtype?” by Tasian et al. 2021
- In addition to ref. 6: “Advances in the Diagnosis and Treatment of Pediatric Acute Lymphoblastic Leukemia” by Inaba et al. 2022.
- In addition to ref. 8 and others on ETV6: “ETV6: a versatile player in leukemogenesis” by Stefan K Bohlander 2005.
- In addition to already mentioned studies at page 4 on “Deletion, mechanism and clinical significance of t(12;21), please add AIEOP-BFM study from Conter et al. Blood 2010, which reported 25.9% ETV6::RUNX1 frequency.
-In addition, the ref. 84 page 12 is too old and not updated, please remove it and add Conter study (Blood 2010).
5) Page 16, paragraph 4, lines 639 and followings: the proposal of gene editing as personalized treatment is not appropriate and it's premature. As far as we are on these techniques and technologies, they are appropriate as a tool to face the disease but not ready for clinical applications. Please smooth this sentence.
Author Response
Response to Reviewer 1 Comments:
Dear Sir or Madam, thank you very much for the review out manuscript entitled: „ The landscape of secondary genetic rearrangements in paediatric patients with B-cell acute lymphoblastic leukaemia with t(12;21 )
In response to your comment, we would like to thank you for appreciating our manuscript.
Thus, you can find paragraphs which involves minor changes in the corrected manuscript.
We provided the following changes:
- The paragraph about treatment and prognosis has been numbered and significantly shortened. We organized the studies in order of prognostic significance at the time of diagnosis, through treatment and lastly with studies on patients with relapse. On page 12, we corrected the sentence about prognostic significance of the results of the study, according to the given advice.
- We reorganized the studies in the 2nd chapter according to the frequency of occurrence, based on Table 1. Now the paragraph is well-balanced and readable.
- In the introduction and chapter 2nd, “second-hit” concept was described in greater detail.
- Suggested references have been added or changed in the article.
We do honestly hope that it will satisfy you and improve the quality of our work. Once again we are very grateful for your review and remain open if you have any other remarks or suggestions that will make our work merit publication in “Cells”
Reviewer 2 Report
The Review article by Kaczmarska and colleagues summarizes current knowledge on secondary chromosomal rearrangements in pediatric B-ALL with ETV6-RUNX1 abnormalities.
The topic is well introduced and of high interest. The review is well structured and mostly concise. All aspects of the primary fusion event are covered and illustrated. A very comprehensive overview on studies assessing secondary genetic events and treatment is given. With some shortening (see below) the article will be of high value for the readership.
Comments:
The authors mention inherited ETV6 mutations in thrombocytopenia but miss mentioning FPD-AML in case of inborn RUNX1 mutations. This should be mentioned.
Focussing on secondary chromosomal rearrangements, the authors may consider reducing the information on ethnical differences in ETV6-RUNX1 occurance or link it to the secondary events.
While the authors present a very comprehensive overview on studies detecting secondary events, it may be helpful for the reader to shorten this paragraph by combining some primary literature with similar findings. The same applies for the treatment section.
Author Response
Response to Reviewer 2 Comments:
Dear Sir or Madam, thank you very much for the review out manuscript entitled: „ The landscape of secondary genetic rearrangements in paediatric patients with B-cell acute lymphoblastic leukaemia with t(12;21 )
In response to your comment, we would like to thank you for appreciating our manuscript.
Thus, you can find paragraphs which involves minor changes in the corrected manuscript.
We provided the following changes
- As suggested, we added a study on FPD-AML.
- In Chapter 1st, Section 1.3, we have arranged the incidence of ETV6::RUNX1 translocations in different countries according to prevalence to make it clearer to the reader. We have reviewed all the publications cited, and in none of them we found information about differences in the incidence of secondary rearrangements between different ethnicities.
- We reorganized the studies in the 2nd chapter according to the frequency of occurrence, based on Table 1. Now the paragraph is well-balanced and readable. The Chapter 3rd, about treatment has been shorted and organized starting with improvements in chemotherapy, through targeted therapies and ending with future methods based on gene editing. The chapter 4 was also shortened.
We do honestly hope that it will satisfy you and improve the quality of our work. Once again we are very grateful for your review and remain open if you have any other remarks or suggestions that will make our work merit publication in “Cells”.
Round 2
Reviewer 1 Report
The authors have made all the required changes and additions. The manuscript has improved.